# Regulation of human mTOR complexes by DEPTOR

**Matthias Wälchli, Karolin Berneiser, Francesca Mangia, Stefan Imseng, Louise-Marie Craigie, Edward Stuttfeld, Michael N Hall, Timm Maier\***

Biozentrum, University of Basel, Basel, Switzerland

**Abstract** The vertebrate-specific DEP domain-containing mTOR interacting protein (DEPTOR), an oncoprotein or tumor suppressor, has important roles in metabolism, immunity, and cancer. It is the only protein that binds and regulates both complexes of mammalian target of rapamycin (mTOR), a central regulator of cell growth. Biochemical analysis and cryo-EM reconstructions of DEPTOR bound to human mTOR complex 1 (mTORC1) and mTORC2 reveal that both structured regions of DEPTOR, the PDZ domain and the DEP domain tandem (DEPt), are involved in mTOR interaction. The PDZ domain binds tightly with mildly activating effect, but then acts as an anchor for DEPt association that allosterically suppresses mTOR activation. The binding interfaces of the PDZ domain and DEPt also support further regulation by other signaling pathways. A separate, substrate-like mode of interaction for DEPTOR phosphorylation by mTOR complexes rationalizes inhibition of non-stimulated mTOR activity at higher DEPTOR concentrations. The multifaceted interplay between DEPTOR and mTOR provides a basis for understanding the divergent roles of DEPTOR in physiology and opens new routes for targeting the mTOR-DEPTOR interaction in disease.

## Introduction

DEP domain-containing mTOR interacting protein (DEPTOR), conserved in vertebrates, modulates the activity of the serine/threonine kinase mammalian target of rapamycin (mTOR), a master regulator of cell growth. mTOR acts in two functionally distinct multiprotein complexes, mTOR complex 1 (mTORC1) and mTORC2 (*Sabatini et al., 1994*; *Jacinto et al., 2004*; *Sarbassov et al., 2004*; *Liu and Sabatini, 2020*; *Loewith and Hall, 2011*; *Loewith et al., 2002*), and DEPTOR is the only protein reported to bind and inhibit both mTOR complexes (*Peterson et al., 2009*).

DEPTOR is a 46 kDa protein comprising an N-terminal DEP (Dishevelled, Egl-10, and Pleckstrin) domain tandem, herein referred to as DEPt, and a C-terminal PDZ (postsynaptic density 95, disks large, zonula occludens-1) domain. The PDZ domain has been suggested to interact with mTOR (*Peterson et al., 2009*), and DEPt mediates phosphatidic acid (PA) binding (*Weng et al., 2021*). The linker connecting DEPt and the PDZ domain contains a phosphodegron motif. mTOR phosphorylates this motif, leading to subsequent additional phosphorylation, ubiquitylation by the SCF$^{\beta TrCP}$ E3 ubiquitin ligase, and DEPTOR degradation (*Gao et al., 2011*; *Zhao et al., 2011*; *Duan et al., 2011*). DEPTOR degradation, in turn, leads to activation of mTORC1 and inactivation of mTORC2 via the mTOR negative feedback loop. OTU domain-containing ubiquitin aldehyde-binding protein 1 (OTUB1) counteracts this process by deubiquitylating DEPTOR (*Zhao et al., 2018*). The interplay of mTOR and DEPTOR with the feedback loop from mTORC1 to mTORC2 and other signaling pathways leads to complex response patterns linked to variations in DEPTOR abundance depending on cell type and state (*Caron et al., 2018*; *Catena and Fanciulli, 2017*).

DEPTOR plays central roles in cancer, obesity, and immunodeficiency (*Caron et al., 2018*; *Laplante et al., 2012*; *Wedel et al., 2019*; *Wedel et al., 2016*; *Caron et al., 2016*). It can act as both an oncoprotein and a tumor suppressor (*Caron et al., 2018*), and its effect in modulating PI3K-AKT signaling is variable depending on cancer type and cellular status. DEPTOR levels are low in most cancers due to

**\*For correspondence:**
timm.maier@unibas.ch

**Competing interest:** The authors declare that no competing interests exist.

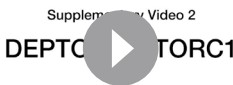

**Video 1.** The DEP domain-containing mammalian target of rapamycin (mTOR) interacting protein (DEPTOR)-mTOR complex 2 (mTORC2). Overview of cryo-EM reconstruction and model, highlighting functionally relevant sites and interactions discussed in the associated manuscript.

https://elifesciences.org/articles/70871/figures#video1

active PI3K signaling (*Caron et al., 2018*). In few cancers, including multiple myeloma (*Peterson et al., 2009*), DEPTOR is overexpressed and promotes cancer cell survival. DEPTOR expression levels are increased in white adipose tissue in obesity and DEPTOR promotes adipogenesis by tuning down mTORC1 feedback control and thereby activating AKT signaling (*Laplante et al., 2012*). Despite its relevance to human health, DEPTOR is the only direct protein regulator of mTOR complexes whose molecular mechanism of action is unknown (*Yang et al., 2017*; *Anandapadamanaban et al., 2019*; *Rogala et al., 2019*).

## Results

To investigate the interplay of DEPTOR and mTOR in both human mTOR complexes, we combined cryo-EM analysis of recombinantly expressed and purified DEPTOR-mTORC2 and DEPTOR-mTORC1 complexes at resolutions of 3.2 and 3.7 Å (*Figure 1a, b and c*; *Figure 1—figure supplements 1–3 Video 1* and *Video 2*), respectively, with crystallographic and in solution structural characterization of the DEPt region of DEPTOR and biochemical analysis. The core architecture of the cryo-EM reconstructions of the two mTOR complexes in association with DEPTOR largely resembles that of their DEPTOR-free states (*Figure 1a, b and c*; *Yang et al., 2017*; *Scaiola et al., 2020*). In the mTORC1 and mTORC2 complexes associated with DEPTOR, the mTOR active site adopts a non-activated conformation (*Yang et al., 2017*; *Scaiola et al., 2020*; *Figure 1—figure supplement 3g*) and is not occupied by substrates. Inositol-hexakis-phosphate (IP6) was recently found to bind to mTORC1 and mTORC2, albeit without clear activating or inhibitory effect (*Scaiola et al., 2020*; *Gat, 2019*), and its binding is undisturbed in DEPTOR-bound mTORC1 and mTORC2 complexes. Binding sites for short linear TOS and RAIP motifs in mTORC1 substrates remain empty in the mTORC1-DEPTOR complex (*Tee and Proud, 2002*; *Nojima, 2003*; *Schalm et al., 2002*). Consistent density for DEPTOR is observed in two regions of the FAT domain of mTOR for both mTORC1 and mTORC2, in agreement with a regulatory effect of DEPTOR on both complexes (*Peterson et al., 2009*).

The DEPTOR-mTOR interaction occurs in two steps. In one step, the DEPTOR PDZ domain binds the mTOR FAT domain. The PDZ domain core (aa$_{DEPTOR}$326–409) adopts a canonical PDZ fold and binds the TRD2 subdomain (*Yang et al., 2013*) of the mTOR FAT domain (*Figure 2a and b*; *Figure 2—figure supplement 1a*). The interaction interface is formed by a conserved surface of the PDZ domain and three mTOR helices (aa$_{mTOR}$1525–1578) (*Figures 1, 2a and b*; *Figure 2—figure supplement 1a,b,c*). The canonical PDZ domain peptide binding groove (*Lee and Zheng, 2010*) is present, but remains unoccupied in the interaction of the DEPTOR PDZ domain with mTOR (*Figure 2—figure supplement 1d*). This opens the possibility that binding of other, yet unknown protein partners via a canonical PDZ-peptide interaction to the DEPTOR PDZ domain could further strengthen or inhibit the mTOR-PDZ interaction. To the best of our knowledge, the mode of interaction of the DEPTOR PDZ domain with mTOR has not been observed for other PDZ domains. The binding interface between mTOR and the DEPTOR PDZ domain is considerably enlarged by contributions from regions which are known or predicted to be disordered in isolated mTOR complexes or DEPTOR. A loop in the Horn (also known as N-HEAT) (*Yang et al., 2017*; *Aylett et al., 2016*) region of mTOR (aa$_{mTOR}$290–350,

**Video 2.** The DEP domain-containing mammalian target of rapamycin (mTOR) interacting protein (DEPTOR)-mTOR complex 1 (mTORC1). Overview of cryo-EM reconstruction and model, highlighting functionally relevant sites and interactions discussed in the associated manuscript.

https://elifesciences.org/articles/70871/figures#video2

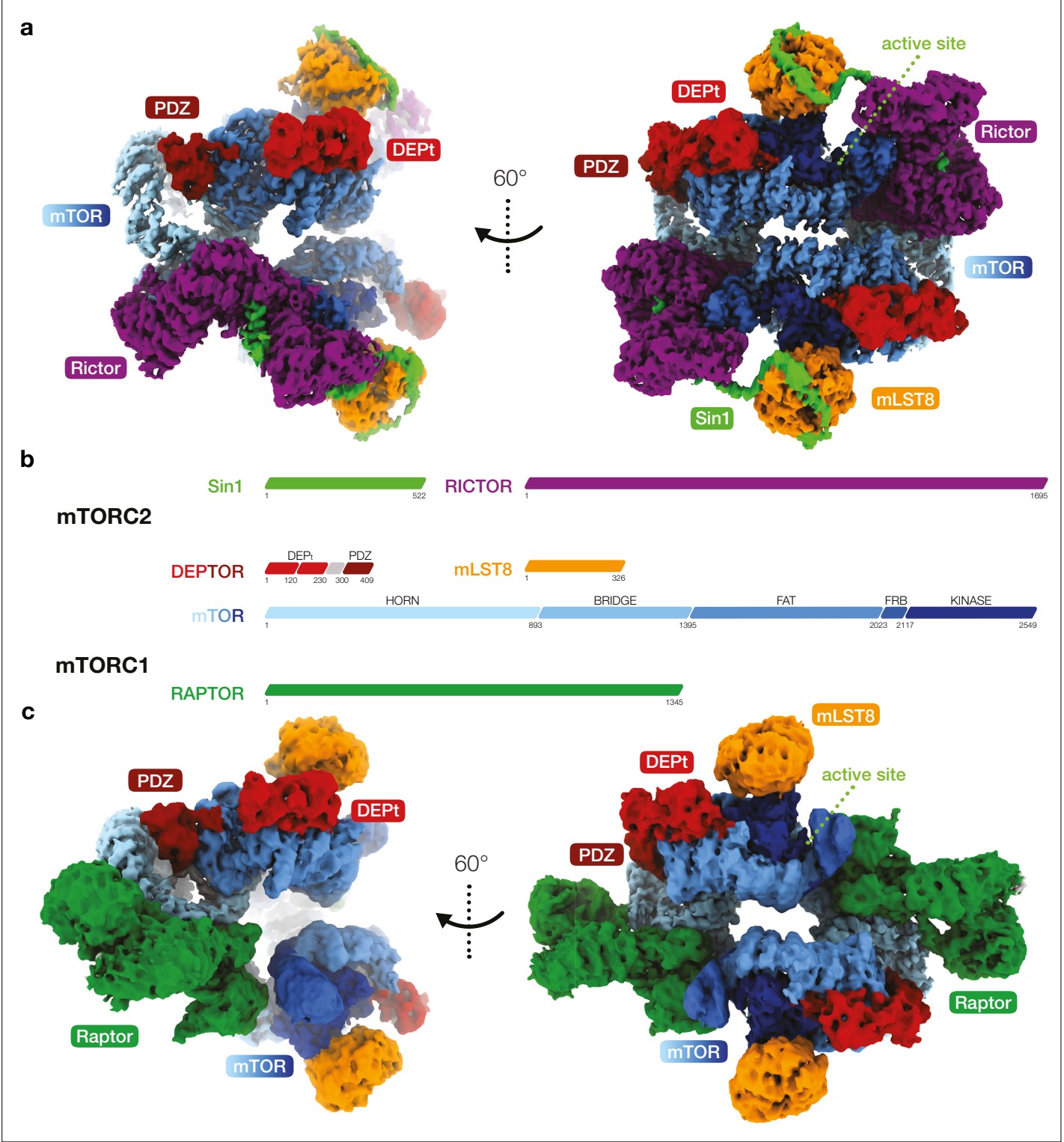

**Figure 1.** Cryo-EM reconstruction of DEP domain-containing mammalian target of rapamycin (mTOR) interacting protein (DEPTOR)-bound mTOR complexes 1 and 2 (mTORC1 and mTORC2). (**a**) Composite map of overall and local focused cryo-EM reconstructions of DEPTOR-bound mTORC2. (**b**) Schematic representation of the domain architecture of mTORC1, mTORC2, and DEPTOR. (**c**) Composite map of overall and local focused cryo-EM reconstructions of DEPTOR-mTORC1. In (**a**) and (**c**) proteins are colored according to the schemes in (**b**). DEPTOR binds to mTORC1 and mTORC2 in virtually identical manner via its extended PDZ-linker and DEP domain tandem (DEPt) regions associating with the FAT domain of mTOR.

The online version of this article includes the following figure supplement(s) for figure 1:

*Figure 1 continued on next page*

*Figure 1 continued*

**Figure supplement 1.** Cryo-EM data processing DEP domain-containing mammalian target of rapamycin (mTOR) interacting protein (DEPTOR)-mTOR complex 2 (mTORC2).

**Figure supplement 2.** Cryo-EM data processing of DEP domain-containing mammalian target of rapamycin interacting protein (DEPTOR)-mTOR complex 1 (mTORC1).

**Figure supplement 3.** Local resolution and active site state of mammalian target of rapamycin (mTOR) complexes in cryo-EM reconstructions.

DEPTOR-binding loop) (*Figure 2c*) was disordered in previous reconstructions of mTOR complexes in the absence of DEPTOR, and its function remained elusive (*Yang et al., 2017*; *Scaiola et al., 2020*). In complex with DEPTOR, residues $aa_{mTOR}304–317$ are ordered and the backbone of residues $aa_{mTOR}290–303$ connecting to the Horn is visible at lower resolution (*Figure 2—figure supplement 1e*). Residues $aa_{mTOR}304–306$ interact with the DEPTOR PDZ domain and residue $F_{mTOR}306$ is inserted between the DEPTOR PDZ and mTOR FAT domains as an integral part of the interface (*Figure 2—figure*

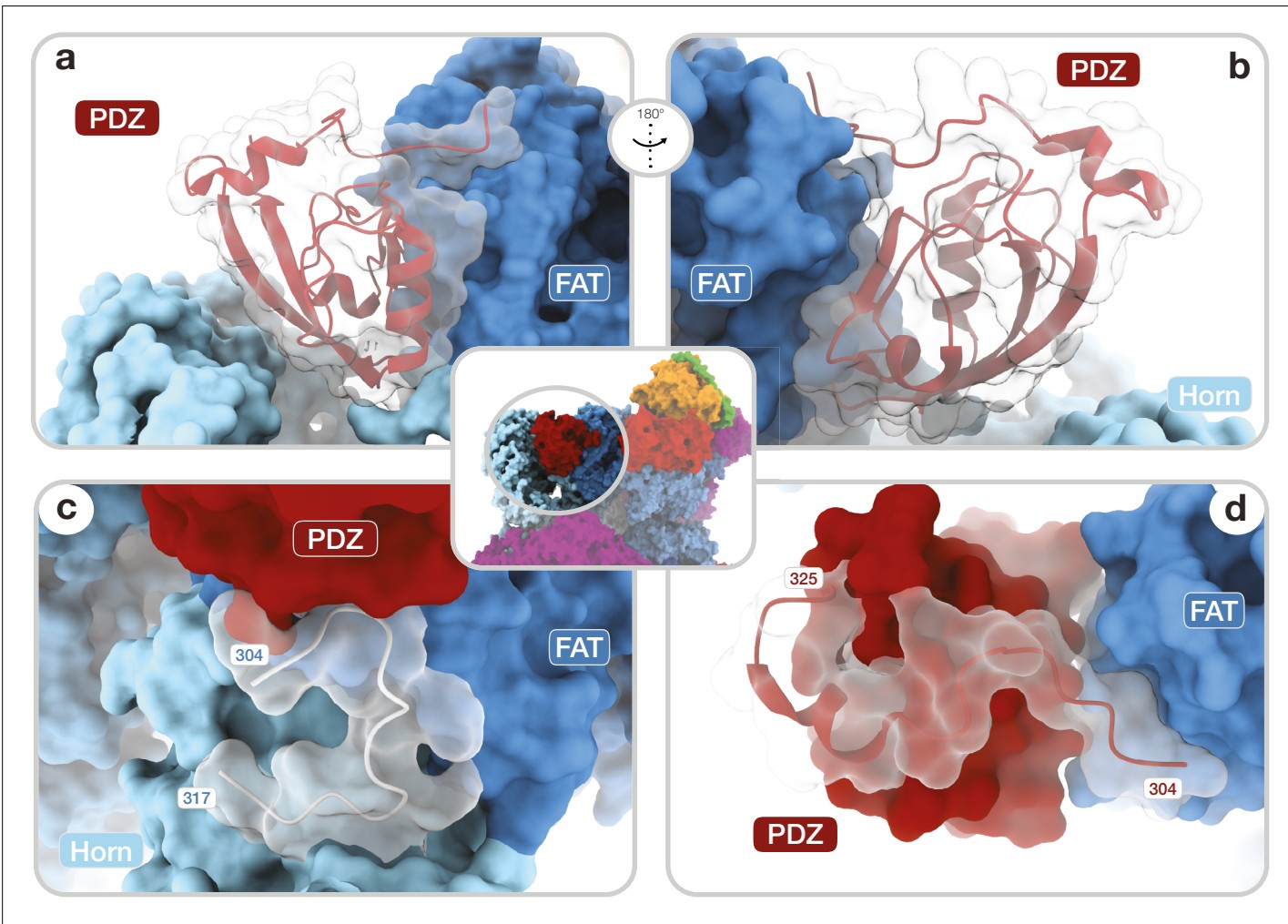

**Figure 2.** Architecture of the DEP domain-containing mammalian target of rapamycin (mTOR) interacting protein (DEPTOR) PDZ domain and its interaction with mTOR. (**a, b**) Front (**a**) and back (**b**) view of DEPTOR PDZ bound to the mTOR FAT domain. The PDZ domain (shown as transparent surface with red cartoon) binds to a hinge in the FAT domain of mTOR. (**c**) The PDZ domain N-terminal extension stretches toward the FAT domain. The adjacent N-terminal linker inserts into a groove on the FAT domain and substantially contributes the PDZ-mTOR interface. (**d**) Loop region ($aa_{mTOR}290–350$) in the mTOR Horn-region (transparent with cartoon) is disordered in free mTOR complexes and contributes to the mTOR-PDZ interface and thereby creates a link between the Horn-region and the FAT domain of mTOR and the DEPTOR PDZ domain.

The online version of this article includes the following figure supplement(s) for figure 2:

**Figure supplement 1.** PDZ domain interaction with mammalian target of rapamycin (mTOR) complexes.

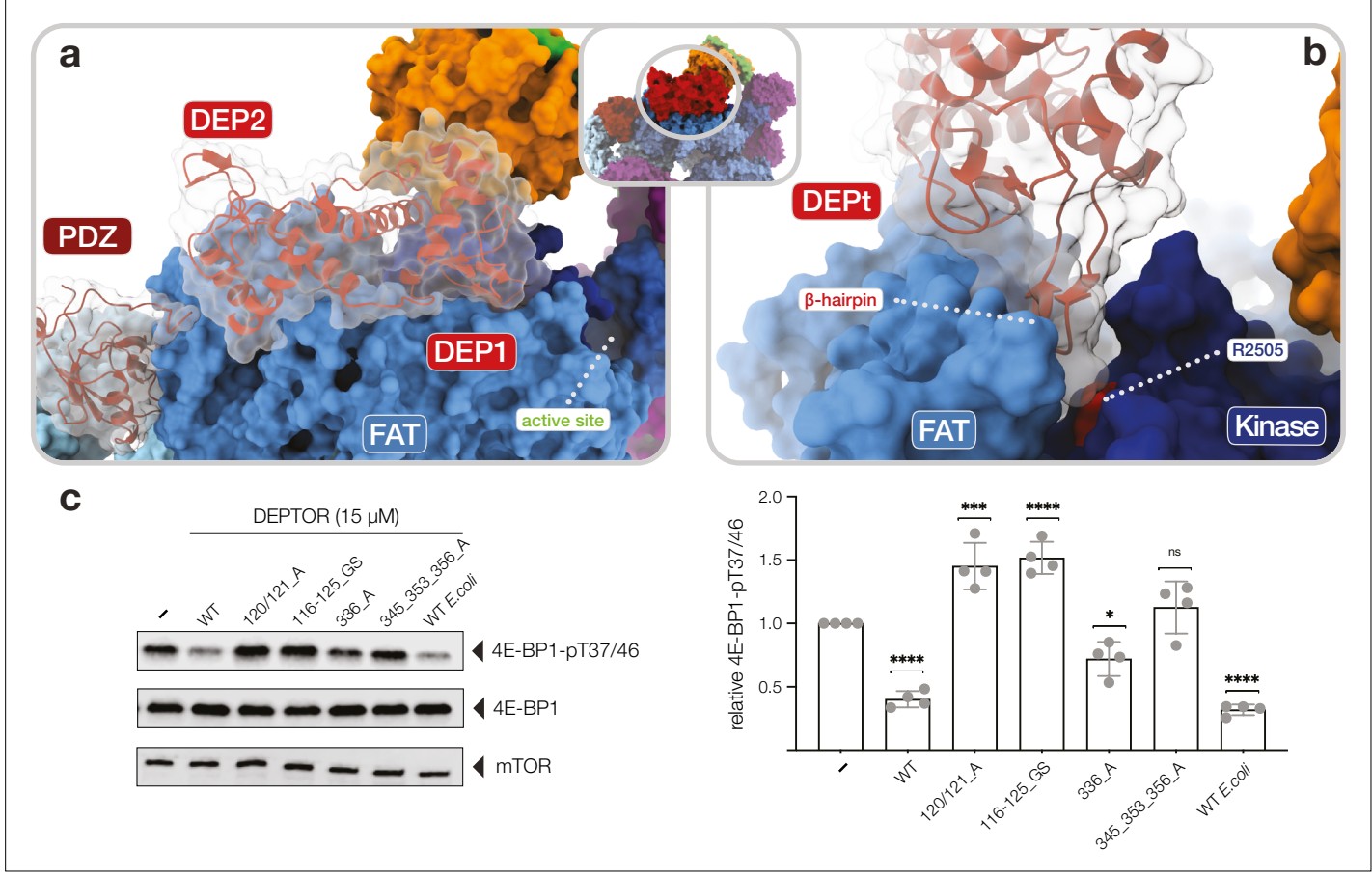

**Figure 3.** Interactions of the DEP domain-containing mammalian target of rapamycin (mTOR) interacting protein (DEPTOR) DEP domain tandem (DEPt) region with mTOR. (**a**) Surface representation of DEPTOR (transparent with cartoon in red) bound to mTOR complex 2 (mTORC2). The DEPt region binds centrally on top of the helical repeats of the FAT domain. (**b**) The protruding hairpin of the first DEP domain of DEPt inserts into a crevice between the kinase and FAT domain of mTOR. The DEPTOR-displacing mutant R2505P (*Grabiner et al., 2014*) is located in close proximity. (**c**) Analysis of the impact of wild-type and mutant forms of DEPTOR on Rheb-stimulated mTORC1 activity. Mutants are described in *Figure 3—figure supplement 2*. mTORC1 was incubated with 4E-BP1 and Rheb for stimulation, in the presence of DEPTOR wild-type and mutants. Reactions were separated by SDS-PAGE and analyzed by western blot. 4E-BP1 phosphorylation was detected with an antibody specific to phosphorylation of residues T37/46. Quantification (mean ± SD) of western blots in 4E-BP1-pT37/46 signals were normalized to total 4E-BP1 signals and the statistical significance of changes between control (0 µM DEPTOR) and DEPTOR variants determined by one-way ANOVA. ****$p < 0.0001$, ***$p < 0.001$, *$p < 0.05$, ns$p > 0.05$, n = 4.

The online version of this article includes the following figure supplement(s) for figure 3:

**Source data 1.** Source data of kinase assay.

**Figure supplement 1.** DEP domain tandem (DEPt) in crystals and associated with mTOR complex 2 (mTORC2).

**Figure supplement 2.** Mutations in PDZ and DEP domain tandem (DEPt) interface.

*supplement 1f*). The DEPTOR PDZ domain together with the DEPTOR-binding loop forms a structural link between the Horn and FAT domain of mTOR, positioned to mediate conformational crosstalk between different subregions of the mTOR complexes. The DEPTOR linker connecting DEPt and the PDZ domain remains largely unresolved (aa$_{DEPTOR}$231–303), and only the C-terminal region of the linker (aa$_{DEPTOR}$ 304–325) is ordered when DEPTOR is bound to mTOR complexes. Residues aa$_{DEPTOR}$309–325 provide an N-terminal extension to the PDZ core domain, while aa$_{DEPTOR}$304–308 bind a groove formed by alpha-helices 14–16 of the mTOR FAT domain (*Figure 2d*). The linker-mTOR interaction enlarges the interface formed by the PDZ core domain suggesting functional relevance of linker residues aa$_{DEPTOR}$304–308 for DEPTOR-mediated regulation of mTOR.

The other step of the DEPTOR-mTOR interaction is mediated by the DEPTOR DEPt region and the mTOR FAT domain (*Figure 1a and c*; *Figure 3a*). The DEPt region bound to mTOR is less well resolved

in overall high-resolution reconstructions as a consequence of local flexibility and partial occupancy. 3D variability analysis, focused classification, and local refinement (*Figure 1—figure supplements 1–S.1a and 2a*) led to clear visualization of the overall fold and individual secondary structure elements (*Figure 3—figure supplement 1a*) at a local resolution of approximately 4–6 Å (*Figure 1—figure supplement 3e,f*). To obtain a pseudo-atomic model of the second DEPTOR-mTOR binding interface, we determined an X-ray crystal structure of a recombinant DEPt region ($aa_{DEPTOR}$ 1–230) at 1.93 Å resolution in a domain-swapped conformation as revealed by small-angle X-ray scattering (SAXS) in solution (*Figure 3—figure supplement 1b,c*). Each of the two domains in DEPt adopts a characteristic DEP domain fold comprising an alpha-helical core and a protruding beta-hairpin arm. In the DEPt, the two DEP domains are interacting via their N-terminal alpha-helices and a C-terminal extension of the second DEP domain that folds back onto the first DEP domain (*Figure 3—figure supplement 1b*). Binding of DEPt to mTOR preserves the overall fold of DEPt, but is linked to a 3.6 Å translation and 39° rotation between the two DEP domains (*Figure 3—figure supplement 1d*).

The DEPt region binds on top of the helical repeats of the mTOR FAT domain in the region $aa_{mTOR}$1680–1814 (*Figure 3a*). The binding interface of DEPt with mTOR is smaller than that of the PDZ-and-linker interaction site (~950 Å$^2$ compared to ~1100 Å$^2$). The N-terminal DEP domain, including the linker to the second DEP domain, forms the major part of the interface (~700 Å$^2$ compared to ~250 Å$^2$) and is better ordered than the C-terminal DEP domain (*Figure 1—figure supplement 3e,f*; *Figure 3—figure supplement 1a*). Notably, the N-terminal DEP domain is absent in one of the two known isoforms of human DEPTOR (*Ota et al., 2004*), likely abolishing DEPt-mTOR association. The protruding beta-hairpin of the N-terminal DEP domain inserts into a crevice between the FAT and kinase domains of mTOR (*Figure 3b*), where residue $R_{mTOR}$2505 is located. This residue is altered to proline in a cancer-associated mutation that weakens DEPTOR binding to mTOR (*Grabiner et al., 2014*; *Sato et al., 2010*) and cannot be compensated by DEPTOR over-expression, underlining the functional relevance of this interaction (*Figure 3b*; *Xu et al., 2016*). mTOR interacting residues of DEPt are highly conserved (*Figure 3—figure supplement 1e*) and the surface electrostatic potentials around the interface are complementary (*Figure 3—figure supplement 1f*). Notably, two positively charged patches in DEPt, which are involved in mTOR interaction, were found to bind PA (*Weng et al., 2021*). PA has been reported to displace DEPTOR from mTOR complexes (*Yoon et al., 2015*).

Previously described mechanisms of mTOR inhibition include ATP-competitive binding to the kinase active site in mTORC1 and mTORC2 (e.g. Torin1) (*Thoreen et al., 2009*), steric hindrance of access to the active site by the FKBP12-Rapamycin complex (*Yang et al., 2017*; *Choi et al., 1996*), and competition with substrate-guiding interactions specific to mTORC1 by the FRB domain binding protein inhibitor PRAS-40. Competitive binding at other substrate recognition elements, such as the TOS and RAIP motif binding sites in mTORC1 (*Yang et al., 2017*; *Tee and Proud, 2002*; *Nojima, 2003*; *Schalm et al., 2002*; *Böhm, 2021*; *Wang et al., 2007*) or C-terminal parts of SIN1 (*Tatebe et al., 2017*) in mTORC2, provides alternative target sites for mTOR inhibition. Recently developed small molecule mTOR inhibitors either utilize the above inhibitory mechanisms or their detailed mode of action is still unknown (*Benavides-Serrato et al., 2017*; *Benjamin et al., 2011*; *Wang et al., 2020*; *Guenzle et al., 2020*).

Notably, DEPTOR is not only a modulator of mTORC1 and mTORC2 activity, but also a substrate of mTOR in mTORC1 and mTORC2 (*Peterson et al., 2009*; *Gao et al., 2011*; *Zhao et al., 2011*; *Duan et al., 2011*). Indeed, we observe weak residual density in the DEPTOR-mTORC1 complex at a binding site for helical peptide segments of substrates and the inhibitor PRAS40 on the FRB domain, which might represent a dynamically interacting segment of DEPTOR or copurified interacting proteins (*Figure 4—figure supplement 1a*). Based on distance constraints, binding of the linker of DEPTOR with an extended helix as in PRAS40 to the FRB site is incompatible with the DEPTOR association to mTOR via its PDZ and DEPt regions; a smaller association patch cannot be ruled out, but would require fully extended surrounding linker regions. Still, it is sterically impossible that all sites for mTOR-mediated phosphorylation in the DEPTOR linker ($aa_{DEPTOR}$ 244, 265, 286, 293, 295, 296, 299; *Peterson et al., 2009*; *Gao et al., 2011*; *Zhao et al., 2011*; *Duan et al., 2011*) could reach the mTOR active site when DEPTOR is associated with mTORC1 via its PDZ and DEPt regions. Thus, a secondary linker-mediated, low-affinity binding mode of DEPTOR (or in trans-phosphorylation without recruiting signal) is required and provides a plausible explanation for the residual signal at the FRB domain.

To test the relevance of DEPTOR interactions in the regulation of mTOR activity, we analyzed phosphorylation of 4E-BP1 by Rheb-activated mTORC1. An equivalent in vitro activity assay with activated mTORC2 has not been described. Insect cell and *Escherichia coli* expressed DEPTOR, which are partially phosphorylated or unphosphorylated, respectively, show a 60–70% inhibition of 4E-BP1 phosphorylation at $T_{4E-BP1}37/46$ (*Figure 3c*, *Figure 3—source data 1*). This inhibition is partially abolished by a single mutation and fully reverted by a triple mutation of the core PDZ interface (*Figure 3c*, *Figure 3—figure supplement 2*). Notably, mutations in the interface between DEPt and mTOR lead to stimulation of 4E-BP1 phosphorylation (*Figure 3c*, *Figure 3—figure supplement 2*), suggesting that DEPt mediates the inhibitory effect of DEPTOR on mTOR complexes, while the binding of DEPt interface mutants only via the PDZ domain even mildly activates Rheb-bound mTORC1 (*Figure 3c*).

## Discussion

Our structural and mutational analyses suggest a model for DEPTOR action on mTOR complexes, in which DEPTOR provides an additional layer of control with the ability to stimulate or inhibit the mTOR complexes (*Figure 4a*). In this model DEPTOR associates with high affinity via its PDZ domain anchor, possibly modulated by other PDZ-binding proteins, followed by lower-affinity association of DEPt based on avidity. DEPTOR partially inhibits mTOR activity by a dominant negative effect of DEPt association or moderately stimulates mTOR activity via the influence of the PDZ domain, if DEPt is prevented from mTOR association by PA binding (*Figure 4b*). A suppression of non-stimulated basal mTORC1 or mTORC2 would only be observed at high concentration of DEPTOR (*Figure 4b*) that result in additional substrate-like binding of DEPTOR to mTORCs.

The isolated DEPt region of DEPTOR has been reported to lack significant binding to mTORC1 and to have no effect on mTORC1 activity, resulting in the hypothesis that DEPt is not involved in controlling mTOR activity in the context of full-length DEPTOR (*Peterson et al., 2009*; *Heimhalt et al., 2021*). However, our data show that DEPt, when anchored via the PDZ domain, binds to a region of the mTOR FAT domain and suppresses allosteric activation of mTOR. Avidity of combined strong PDZ and weak DEPt interactions is supported by the earlier observation that full-length DEPTOR inhibits mTORC1 activation already at a lower concentration than required for binding of the isolated PDZ domain: The isolated PDZ domain binds with a $K_d$ of 0.6 μM to the mTORC1 variant $A_{mTOR}1459P$ that mimics activation by Rheb, but the $IC_{50}$ value for DEPTOR in the same system is 30–50 nM (*Heimhalt et al., 2021*). The functional relevance of a similar interplay of strong and extremely weak association has recently been demonstrated for two mTORC1-binding motifs in 4E-BP1, the high affinity TOS motif and the very low-affinity RAIP motif (*Böhm, 2021*).

We observed an unexpected, weak activation of Rheb-stimulated mTORC1 activity in mutants of the DEPt interface which we attributed to an effect of PDZ domain binding. This effect is consistent with the observation of increased 4E-BP1 phosphorylation by mTORC1 in the presence of equimolar isolated PDZ domain (at overall nanomolar concentrations, Figure 2—figure supplement 1/Figure 6—figure supplement 1 in *Heimhalt et al., 2021*). It has also been reported that the PDZ domain has an approximately 10-fold higher affinity ($K_d$ 0.6 μM vs. 7 μM) for binding to activated vs. non-activated mTORC1 (*Heimhalt et al., 2021*), despite a lack of differences in the interface in static structures of activated and non-activated mTOR complexes (*Yang et al., 2017*). Together, these data suggest that association of the PDZ domain is linked to changes in the dynamics of its binding site on mTOR, which are allosterically coupled to mTOR activation. Binding of other proteins to DEPTOR based on a canonical peptide-PDZ domain interaction via the empty PDZ-peptide binding groove may modulate the affinity of the PDZ to mTOR or even its effect on activity when bound to mTOR.

Why are low concentrations of DEPTOR inhibiting Rheb-stimulated mTORC1 but much higher concentrations of DEPTOR are required (*Heimhalt et al., 2021*) for the reported inhibition of non-stimulated mTORCs (*Peterson et al., 2009*; *Gao et al., 2011*; *Laplante et al., 2012*; *Yoon et al., 2015*)?

Facilitated by PDZ binding to mTOR, DEPt associates with non-activated mTOR complexes without inducing structural changes in the FAT region, as visualized here for mTORC1 and mTORC2. We suggest that DEPt association with a region of the FAT domain, that transduces allosteric activation by Rheb, specifically suppresses the conformational coupling between the Rheb binding site and the kinase site and consequently reduces only the stimulation of mTORC1 activity. This may occur by increasing the population of a state of the FAT region that is less competent for transmitting allosteric

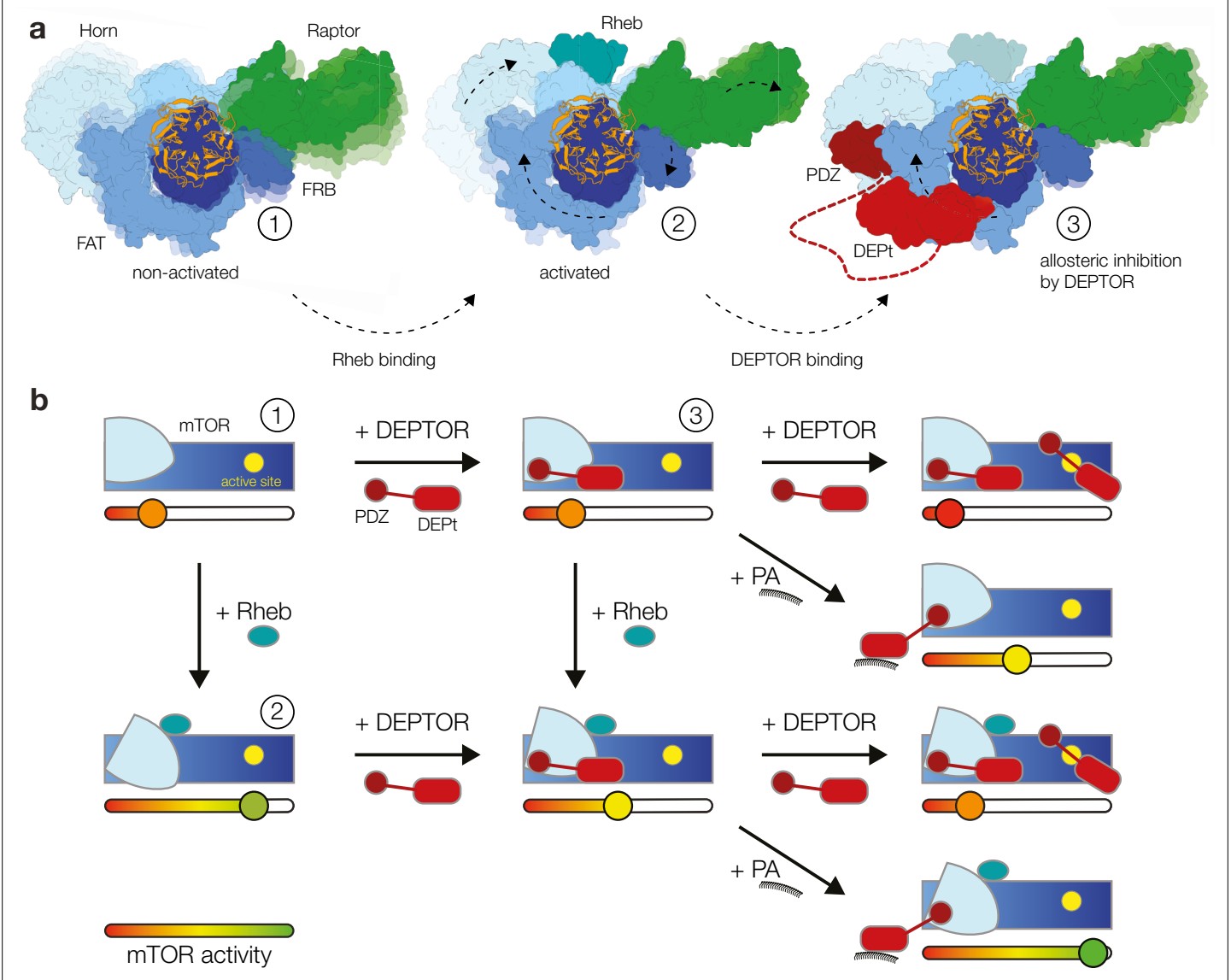

**Figure 4.** Model for the DEP domain-containing mammalian target of rapamycin (mTOR) interacting protein (DEPTOR)-mediated regulation of mTOR activity. (**a**) Structure-based representation of (1) the basal state of non-activated mTOR complex 1 (mTORC1) (based on PDB: 6BCX), (2) the allosteric activation of mTORC1 by Rheb binding (based on PDB: 6BCU), and (3) the impact of DEPTOR association via the PDZ domain and DEP domain tandem (DEPt) on the conformational state and activity of mTORC1. Possible transitions in subpopulations of conformational states are indicated by shadowing. (**b**) Schematic diagram of the suggested regulatory interactions between DEPTOR and mTOR complexes. Structurally characterized states shown in (**a**) are indicated by numbers. DEPTOR binding via the PDZ domain and DEPt prevents allosteric activation. At high concentrations, DEPTOR binds to mTORC1 in a secondary binding mode as a substrate and sterically influences access of other substrates to the active site. Phosphatidic acid (PA) may interfere with the DEPt-mTOR association, relieving the allosteric inhibition of mTORCs. The remaining bound PDZ domain mildly stimulates kinase activity in activated and non-activated mTOR complexes.

The online version of this article includes the following figure supplement(s) for figure 4:

**Figure supplement 1.** Residual density in a substrate recruitment site at the FRB domain for DEP domain-containing mammalian target of rapamycin (mTOR) interacting protein (DEPTOR)-bound mTOR complex 1 (mTORC1).

activation, either directly the state observed in non-activated mTOR or other intermediate states. Such a mode of action suggests the absence of inhibition of non-stimulated basal mTORC activity at low concentrations of DEPTOR. An alternative explanation for the lack of inhibition at low DEPTOR concentrations (*Heimhalt et al., 2021*) could be the failure to associate with mTORCs as a result of the differential interaction of the PDZ domain with stimulated vs. non-stimulated mTORC1, $K_d$ of 0.6 µM

vs. 7 µM (*Heimhalt et al., 2021*), respectively. However, based on our demonstration of an additional interface for DEPt binding, the avidity of combined DEPt and PDZ association suggests that the association of full-length DEPTOR still occurs at a concentration lower than the approximately 15–50 µM required for effective inhibition of non-stimulated mTORC1 (*Heimhalt et al., 2021*).

A plausible explanation for inhibition of mTORC1 and mTORC2 at higher concentrations could be a secondary, lower-affinity binding mode at excess concentrations of DEPTOR over mTOR that does not involve interactions of PDZ and DEPt with mTOR. Indeed, DEPTOR is a substrate for mTORC1 and mTORC2 (*Peterson et al., 2009*; *Gao et al., 2011*; *Zhao et al., 2011*; *Duan et al., 2011*). Substrate recruitment by TOR complexes involves specific substrates recruitment via medium- and low-affinity interactions outside the kinase domain for many substrates (*Yang et al., 2017*; *Tee and Proud, 2002*; *Nojima, 2003*; *Schalm et al., 2002*; *Böhm, 2021*; *Tatebe et al., 2017*), but the primary DEPTOR binding via PDZ domain and DEPt is not suitable for recruitment, suggesting lower-affinity secondary binding modes to mTORC1 and mTORC2. Indeed, such an alternative, substrate-like weak interaction is indicated by residual density observed here in a known substrate recruitment site for mTORC1. We have previously shown that the core substrate recognition regions in mTORC2 are flexibly disposed, explaining why substrate interactions are not visualized in the current type of cryo-EM analysis (*Scaiola et al., 2020*).

Substrate-like association of DEPTOR, via the FRB domain or other regions in mTORC2, and simultaneous recruitment of other substrates with their respective binding motifs, for example, 4E-BP1 with its TOS and RAIP motifs, would result in a mutual restriction of access to the mTOR active site that may partially be uncoupled from solution concentrations and dominated by local protein dynamics. At the same time, we consider that the core mechanism for the high-affinity mode of inhibition of activated mTOR complexes by DEPTOR is unlikely to be based on a PRAS40-like FRB interaction because (1) the binding site on the FRB domain is not accessible in mTORC2 (*Figure 4—figure supplement 1b*), (2) this would leave the conserved characteristic DEPt and its mTOR interaction involving $R_{mTOR}2505$ without assigned function, and (3) it provides no additional explanation for differential effects on stimulated and non-stimulated mTORC1, as the binding site on the FRB domain is not coupled to allosteric activation.

Notably, the interface of DEPt and mTOR suggested here to mediate inhibition of mTOR inhibition involves regions of DEPt that have been recently implicated in PA interaction (*Weng et al., 2021*). We hypothesize that DEPt interaction with PA may control DEPt association with mTOR, resulting in either PDZ-based activation or DEPt-based down-regulation of activated mTOR complexes. This would create a mechanistic link between PA signaling and mTOR activation on membranes (*Figure 4b*; *Takahara et al., 2020*).

DEPTOR has been characterized as a modulator of mTOR activity with a profound impact on metabolism and cancer (*Liu and Sabatini, 2020*). However, its divergent and orthogonal effects on cell physiology, including its apparently antagonistic roles as an oncoprotein and tumor suppressor, have remained enigmatic. Here, we provide a structure-guided model of the complex interplay of DEPTOR with mTORC1 and mTORC2 that identifies orthogonal contributions by different interacting regions of DEPTOR, and further potential for modulation by crosstalk from other signaling pathways. The molecular insights provided here will be a crucial component for targeted dissection of DEPTOR effects on mTOR signaling to further understand the divergent effects of DEPTOR in physiology and disease.

## Materials and methods
### Protein expression and purification

Sf21 insect cells (Expression Systems) were grown in HyClone insect cell media (GE Life Sciences) and baculovirus was generated according to *Fitzgerald et al., 2006*. mTORC2 was expressed and purified as previously described with an internal FLAG-tag inserted after D258 (*Scaiola et al., 2020*). Purified mTORC2 was concentrated in gel filtration buffer, supplemented with 5% w/v glycerol and stored at –80 °C until further use.

For expression of human mTORC1, Sf21 insect cells were infected with baculovirus as described previously (*Aylett et al., 2016*). Cells were harvested 72 hr after infection by centrifugation at 800×*g* for 15 min and stored at –80 °C until further use. Cell pellet was lysed in 50 mM bicine (pH 8),

250 mM NaCl, and 5 mM MgCl$_2$ using a dounce homogenizer and the lysate was cleared by ultracentrifugation. Soluble protein was incubated with 7 ml of anti-DYKDDDDK agarose beads (Genscript, Piscataway, NJ) for 1 hr at 4 °C. The beads were transferred to a 50 ml gravity flow column (Bio-Rad) and washed four times with 200 ml of wash buffer containing 50 mM bicine (pH 8), 150 mM NaCl, and 5 mM MgCl$_2$. Protein was eluted by incubating beads for 60 min with 10 ml of wash buffer supplemented with synthetic DYKDDDDK peptide (0.6 mg/ml) (Genscript, Piscataway, NJ). The eluate was combined with three additional elution steps using synthetic DYKDDDDK peptide (0.1 mg/ml) and 5 min incubation time. The eluted protein was concentrated using a 100,000 Da molecular mass cutoff centrifugal concentrator with regenerated cellulose membrane (Amicon) and purified by size exclusion chromatography on a custom Superose 6 Increase 10/600 GL gel filtration column equilibrated with 25 mM bicine (pH 8.0), 200 mM NaCl, 5 % glycerol, and 2 mM tris(2-carboxyethyl) phosphine (TCEP). Purified mTORC1 was concentrated in gel filtration buffer and stored at –80 °C until further use.

Full-length human wild-type (WT) DEPTOR coding sequence was amplified from pRK5 FLAG human DEPTOR, which was a gift from David Sabatini (Addgene plasmid no 21334) (**Peterson et al., 2009**), and was cloned into a pAceBAC2 expression vector (Geneva Biotech, Geneva, Switzerland) with an N-terminal His10-Myc-FLAG tag by Gateway cloning. For expression of human WT DEPTOR, Sf21 cells were infected with baculovirus, harvested 72 hr after infection by centrifugation at 800×$g$ for 15 min and stored at –80 °C until further use. The cell pellet was lysed in lysis buffer (50 mM PIPES [pH 6.8], 500 mM NaCl, 50 mM imidazole, 2 mM MgCl$_2$, and 2 mM TCEP) using sonication, and the lysate was cleared by ultracentrifugation. The cleared lysate was loaded onto a 25 ml Ni-column (Genscript High Affinity Ni-charged resin), washed with 10 column volumes (CV) of lysis buffer containing 1 mM ATP and eluted with a linear gradient (5CV) to lysis buffer containing 500 mM imidazole. The tag was cleaved overnight using TEV protease, followed by an orthogonal Ni-column (25 ml, Genscript High Affinity Ni-charged resin) to remove Tag and the tagged TEV protease from the sample. The protein was subjected to a final gel filtration chromatography step in 20 mM PIPES (pH 6.8), 150 mM NaCl, 5 % glycerol, and 1 mM TCEP. DEPTOR containing fractions were concentrated to 10 mg/ml and stored at –80 °C until further use.

DEPt (aa$_{DEPTOR}$1–230) was cloned into the vector pETG-10A and expressed in *E. coli* BL21DE3 cells. Cells were grown in 2×YT medium at 37 °C. At an OD$_{600}$ of 0.8, isopropyl-β-D-thiogalactoside (IPTG) was added to a final concentration of 0.75 mM. The cells were further grown for 4–6 hr and harvested by centrifugation. Cells were lysed by sonication in 50 mM HEPES (pH 8), 250 mM NaCl, 40 mM imidazole, 1 mM TCEP, and the lysate was cleared by ultracentrifugation. The cleared lysate was loaded onto a 5 ml Ni-column (Genscript High Affinity Ni-charged resin), washed with 13CV of lysis buffer and eluted with a linear gradient over 5CV to 50 mM HEPES (pH 8), 150 mM NaCl, 500 mM imidazole, and 1 mM TCEP. The tag was cleaved overnight using TEV protease, followed by an orthogonal Ni-column Ni-column (5 ml, Genscript High Affinity Ni-charged resin) to remove Tag and tagged TEV protease from the sample. The protein was subjected to a final gel filtration chromatography step in 20 mM HEPES (pH 8), 150 mM NaCl, 5 % glycerol, and 1 mM TCEP. DEPt containing fractions were concentrated to 14.9 mg/ml and stored at –80 °C until further use.

DNA coding for human WT Rheb (aa$_{Rheb}$ 1–171) (GenBank: D78132) was synthesized by Genscript and cloned into the vector pETG30A coding for a His$_6$-GST-Tag for expression. *E. coli* SoluBL21 cells were grown in ZY medium at 37 °C. At an OD$_{600}$ of 0.65, IPTG was added to a final concentration of 0.5 mM. The cells were further grown for 8 hr and harvested by centrifugation. Cells were lysed using a French Press in 50 mM bicine (pH 8), 500 mM NaCl, 5 mM MgCl$_2$, and 5 mM beta-mercaptoethanol (bME), and the lysate was cleared by ultracentrifugation. The cleared lysate was loaded onto a 5 ml His-Trap HP (GE Healthcare), washed 20CV with lysis buffer supplemented with 20 mM imidazole, and eluted with a linear gradient (5CV) to 50 mM Bicine (pH 8), 250 mM NaCl, 500 mM imidazole, 5 mM MgCl$_2$, and 5 mM bME. The tag was cleaved overnight using TEV protease, followed by an orthogonal Ni-column Ni-column (5 ml His-Trap HP [GE Healthcare]) to remove Tag and tagged TEV protease from the sample. The protein was subjected to a final gel filtration chromatography step (HiLoad 16/600 Superdex 75 pg, Cytiva) in 10 mM bicine (pH 8), 150 mM NaCl, and 1 mM TCEP. Rheb containing fractions were collected, concentrated with a 3000 Da molecular mass cutoff centrifugal concentrator (Amicon), and supplemented with 5 % w/v glycerol and stored at –80 °C until further use.

## In vitro mTORC1 activity assays

mTORC1 kinase activity assays were performed in final concentrations of 50 mM HEPES (pH 7.5), 75 mM NaCl, 2.4 % glycerol, 6 mM $MgCl_2$, and 1 mM TCEP. 4E-BP1 was expressed and purified as previously described (*Böhm, 2021*). For Rheb-activated mTORC1, Rheb was loaded with 2 mM GTPγS (Jena Biosciences) and 5 mM EDTA for 1 hr at room temperature and locked by the addition of $MgCl_2$ to 10 mM final concentration. For the activity assays, 2 nM purified mTORC1 were mixed with 15 µM DEPTOR (WT or mutants), 5 µM Rheb and 320 nM 4E-BP1 as the substrate. Reactions were started by the addition of 1 mM ATP, incubated for 10 min at room temperature, and quenched with 5 × SDS sample buffer. Samples were resolved by SDS-PAGE and transferred onto 0.2 µM pore size nitrocellulose membranes via the Trans-Blot Turbo Transfer System (Bio-Rad). Signals were detected by the LI-COR Fc system (LI-COR Biosciences) using the following antibodies: mTOR (1:1000, RRID:AB_2105622), 4E-BP1-p(T37/46) (1:1000, RRID: AB_560835), 4E-BP1 (1:1000, RRID: AB_331692), and IRDye 800CW goat anti-rabbit IgG (1:17.500, RRID: AB_621843). All antibodies were diluted into an equal mix of TBST and LI-COR intercept (TBS) blocking buffer. Statistical analysis was performed by GraphPad Prism (RRID:SCR_002798), using one-way ANOVA.

## Crystallization, X-ray data collection, and structure determination

The DEPt region ($aa_{DEPTOR}$ 1–230) was crystallized at 4 °C using the sitting-drop vapor diffusion method at a protein concentration of 14.9 mg/ml and using a reservoir (12% w/v PEG 3350, 0.2 M NaCl) to protein ratio of 1:1 in a total drop volume of 0.4 µl. Crystals were transferred into cryoprotectant (reservoir solution with added ethylene glycol to 20% v/v) and vitrified in liquid nitrogen. Crystallographic data were collected at the Swiss Light Source (Paul Scherrer Institute) beamline X06SA at 100 K using an Eiger16M detector (Dectris). Data were collected at a wavelength of 1.0 Å with an exposure time of 0.1 s, a rotation angle of 0.25° for 240° and a detector distance of 299.7977 mm.

Data were processed with DIALS and scaled using aimless (RRID:SCR_015747) (*Evans and Murshudov, 2013*). The structure was solved by molecular replacement using the crystal structure of 4F7Z.pdb (*White et al., 2012*) as search model with the program PHASER (RRID:SCR_014219) (*McCoy et al., 2007*). Model building was done with COOT (RRID:SCR_014222) (*Emsley and Cowtan, 2004*) and the structures were refined with PHENIX (RRID:SCR_014224) (*Adams, 2002*). MolProbity (RRID:SCR_014226) (*Williams et al., 2018*) was used to validate the model. Data and refinement statistics are summarized in *Supplementary file 3*. The final model contains two chains with residues 20–230 of DEPTOR, 19 residues at the N-terminus were not resolved, presumably due to flexibility.

## SAXS data collection and analysis

SAXS data of purified DEPt in gel filtration buffer at 2–14 mg/ml was collected in batch-mode experiments on the B21 beamline at Diamond Light Source (DLS), UK. Solution scattering of the dimeric crystal structure, one monomer of the crystal structure and the DEPt model based on the cryo-EM reconstruction, was evaluated and fitted to the experimental scattering curves using PRIMUS and CRYSOL (*Svergun et al., 1995*; *Manalastas-Cantos et al., 2021*). Fits generated from CRYSOL were plotted using GraphPad Prism version 9.1.0.

## Cryo-EM sample preparation and data collection

Freshly thawed mTORC2 aliquots were mixed with freshly thawed DEPTOR aliquots in 1:8 molar ratio and dialyzed in 20 mM bicine (pH 7.5), 100 mM NaCl, 5 mM $MgCl_2$, 2 mM TCEP, and 0.25 % glycerol before preparing grids. For each grid 4 µl of the sample at 1.2 mg/ml were applied to a Quantifoil R2/2 holey carbon copper grid (Quantifoil Micro Tools), which was mounted in a Vitrobot (Thermo Fisher Scientific) whose chamber was set to 4 °C and 100 % humidity. The grid was immediately blotted with a setting of 2.5–4 s blotting time and rapidly plunge-frozen in liquid ethane. Data were collected using a Titan Krios (Thermo Fisher Scientific FEI) transmission electron microscope equipped with a K2 Summit direct electron detector (Gatan) using SerialEM (RRID:SCR_017293) (*Evans and Murshudov, 2013*; *Supplementary file 1*) in counting mode. During data collection, the defocus was varied between –1 and –3 µm and five exposures were collected per holes. Stacks of frames were collected with a pixel size of 1.058 Å/pixel and a total dose of about 50 electrons/$Å^2$.

Freshly thawed mTORC1 aliquots were mixed with freshly thawed DEPTOR aliquots in 1:12 molar ratio and dialyzed in 20 mM bicine (pH 7.5), 100 mM NaCl, 5 mM $MgCl_2$, 2 mM TCEP, and

0.25 % glycerol before preparing grids. For each grid 4 µl of the sample at 1.4 mg/ml were applied to a Quantifoil R2/2 holey carbon copper grid (Quantifoil Micro Tools), which was mounted in a Vitrobot (Thermo Fisher Scientific) whose chamber was set to 4 °C and 100 % humidity. The grid was blotted after 10 s incubation with a setting of 2.5–4 s blotting time and rapidly plunge-frozen in liquid ethane.

Two datasets were collected using a Glacios (Thermo Fisher Scientific FEI) transmission electron microscope equipped with a K3 direct electron detector (Gatan) using SerialEM (*Mastronarde, 2005*; *Supplementary file 2*) in correlated double sampling mode. During data collection, the defocus was varied between –1 and –2.5 µm. Stacks of frames were collected with a super-resolution pixel size of 0.556 Å/pixel and a total dose of about 50 electrons/Å$^2$.

## Cryo-EM data processing

The DEPTOR-mTORC2 dataset, consisting of 7371 micrographs, was corrected for beam-induced drift using Patch Motion, and the contrast transfer function (CTF) parameters for each micrograph were determined using Patch CTF in cryoSPARC (RRID:SCR_016501) (*Punjani et al., 2017*). After curation of micrographs, we selected 6924 micrographs for further processing. Particles were picked with the blob picker function in cryoSPARC and subjected to reference-free 2D classification. 2D classes showing structural features were used as templates for particle picking using the template picker in cryoSPARC. Iterative 2D classification was used to sort particles. Particle coordinates were transferred to Relion (RRID:SCR_016274) and particles were extracted from micrographs, which have been previously motion-corrected using Relion (*Zivanov et al., 2018*) own implementation of MotionCor2 (RRID:SCR_016499) (*Zheng et al., 2017*) and the CTF estimated using CTFFIND4.1 (RRID:SCR_016732) (*Rohou and Grigorieff, 2015*). An initial 3D autorefinement yielded a reconstruction at 4.03 Å. Particles were subjected to iterative CTF refinement and Bayesian particle polishing (*Zivanov et al., 2020*). Polished particles were imported into cryoSPARC followed by a homogeneous refinement. To prune the dataset, particles were classified by heterogeneous refinement in three classes. The most populated class was selected and particles used in a non-uniform homogeneous refinement followed by an local refinement using a global mask yielding a reconstruction at 3.41 Å (map 1; *Figure 1—figure supplement 1a*; *Punjani et al., 2020*). By subsequent local refinements focusing on one protomer respectively, we obtained reconstructions at 3.27  and 3.32 Å. Particles of the global refinement were symmetry-expanded based on the C2 symmetry of mTORC2. A local non-uniform refinement focused on one protomer yielded a map at 3.16 Å. To reduce heterogeneity, particles were classified without alignment in five classes in Relion. After separate local refinement of these classes in cryoSPARC, four classes were selected. On these particles, signal subtraction of one protomer was performed followed by a local non-uniform refinement focused on the remaining protomer. A final protomer map at a resolution of 3.2 Å was obtained (map 2; *Figure 1—figure supplement 1a*). Due to partial occupancy and high flexibility the DEPt region remained poorly resolved in these reconstructions. Sorting for this region was done by 3D classification without alignment into five classes in Relion using the symmetry-expanded particle stack and a mask for the DEPt region. Only one class showed structural features. Partial signal subtraction of one protomer followed by focused local non-uniform refinement yielded a reconstruction at 3.7 Å (map 3; *Figure 1—figure supplement 1a*). A processing scheme can be found in *Figure 1—figure supplement 1a*.

For the DEPTOR-mTORC1 dataset, micrographs from two data collections were corrected separately for beam-induced drift using patch motion, and the CTF parameters for each micrograph were determined using Patch CTF in cryoSPARC (*Punjani et al., 2017*). We selected a total of 8604 movies after manual curation for further processing. Particles were picked with the blob picker function in cryoSPARC and subjected to reference-free 2D classification. 2D classes showing structural features were used as templates for particle picking using the template picker in cryoSPARC. The particle stack was cleaned by iterative 2D classification. Extracted particles from both datasets were subjected to an initial non-uniform refinement with an ab initio reconstruction as starting model. To sort for heterogeneity, the particles were sorted into three classes using heterogeneous refinement. Particles of the two most populated classes representing a 'wide' and 'tight' mTORC1 conformation were used in a non-uniform homogeneous refinement followed by an local refinement using a global mask (*Punjani et al., 2020*). This yielded a reconstruction of DEPTOR-mTORC1 at 4.07 Å (map 4; *Figure 1—figure*

*supplement 2a*). Subsequent non-uniform local refinements focusing on individual protomers resulted in reconstructions at 3.97 and 3.99 Å. Symmetry expansion was performed on the particles of the overall refinement utilizing the C2 symmetry of the complex. Using non-uniform local refinement of symmetry-expanded particles focused on one protomer, a final map at 3.67 Å was obtained (map 5; *Figure 1—figure supplement 2a*). Sorting for occupancy of the DEPt region was achieved by 3D variability analysis using five principle modes (*Punjani and Fleet, 2021*). Particles were classified in five clusters based on the principle modes. Non-uniform local refinement of the cluster with highest DEPt occupancy resulted in a reconstruction at 4.24 Å (map 6; *Figure 1—figure supplement 2a*). A processing scheme can be found in *Figure 1—figure supplement 2a*.

## Cryo-EM model building and refinement

One protomer of the cryo-EM structure of mTORC2 (PDB: 6ZWM *Scaiola et al., 2020*) was used as initial model and was rigid body fitted into map 2 (*Figure 1—figure supplement 1a*). Minor adjustment of the manual to fit the density was done using COOT (*Emsley and Cowtan, 2004*). The linker of mTOR, which becomes structured upon DEPTOR binding, was identified based on continuous density in maps filtered to lower resolution connecting it to modeled parts of mTOR, but only the better-ordered residues $aa_{mTOR}$ 304–317 were built into full-resolution maps de novo using COOT. Initial models and secondary structure definitions for the PDZ domain were generated using trRosetta and Robetta (*Yang et al., 2020*) (RRID:SCR_021181, RRID:SCR_018805) (*Figure 2—figure supplement 1a*). These initial models were rigid body fitted into the map and adjusted in COOT to fit the density. The N-terminal PDZ extension and following linker binding mTOR ($aa_{DEPTOR}$ 325–304) were built de novo using COOT, guided by continuous density for this region visualized in softened or low-pass filtered maps with higher disorder reducing side chain visibility. The structure was finally real-space-refined using phenix.real_space_refine (*Afonine, 2018*). The resulting structure was rigid body fitted into map 3 (*Figure 1—figure supplement 1a*). One monomer of the DEPt crystal structure was fitted into the extra density. Using the cryo-EM reconstruction and the second monomer of the crystal structure as template, the domain-swapped crystal structure was un-swapped in silico to obtain a physiologically relevant DEPt monomer model. Individual domains were fitted into the map and minor adjustments were carried out manually in COOT. The final model was real-space-refined using phenix.real_space_refine.

For model building of DEPTOR-mTORC1, one protomer of pdb:6BCX (*Yang et al., 2017*) was used as initial model. Individual proteins were rigid body fitted into map 5 (*Figure 1—figure supplement 2a*), keeping residues with unassigned identity as in the higher resolution 6BCX. The model of the PDZ domain, obtained from the DEPTOR-mTORC2 reconstruction, was rigid body fitted into the density. The final model was real-space-refined using phenix.real_space_refine. The obtained model was rigid body fitted into map 6 (*Figure 1—figure supplement 2a*). Additionally, the DEPt model, obtained from the respective DEPTOR-mTORC2 reconstruction, was rigid body fitted into map 6 (*Figure 1—figure supplement 2a*) to yield an mTORC1 protomer with PDZ domain and DEPt bound. The final model was real-space-refined using phenix.real_space_refine. To obtain a model for the mTORC1 dimer with PDZ domain bound, two copies of the PDZ-bound protomer were rigid body fitted into map 4 (*Figure 1—figure supplement 2a*) followed by real-space-refinement using phenix.real_space_refine. All models were validated using phenix and MolProbity (*Williams et al., 2018*).

## Structural analysis and figure generation

Properties of individual protein interfaces between DEPTOR and mTOR were analyzed using PISA (RRID:SCR_015749)(*Krissinel and Henrick, 2007*). To analyze sequence conservation of DEPTOR, we aligned 136 full-length DEPTOR sequences with Clustal Omega (RRID:SCR_001591) (*Madeira, 2019*). The final alignment was used as input for AL2CO (*Pei and Grishin, 2001*) to map conservation onto the DEPTOR structure using ChimeraX (RRID:SCR_015872) (*Pettersen et al., 2021*). To analyze motion of the DEP2 of DEPt induced by binding to mTOR, the DEPt crystal and cryo-EM structure were superimposed onto DEP1. Rotation and translation of DEP2 was analyzed using the PSICO extension of Pymol (RRID:SCR_000305) (*Schrodinger, LLC, 2015*). All density and structure representations, and the movies were generated using UCSF ChimeraX (*Pettersen et al., 2021*). Local resolution was estimated using cryoSPARCv3 (Structura Biotechnology Inc).

## Acknowledgements

We thank T Sharpe at the Biophysics facility and A Schmidt at the Proteomics Core Facility of Biozentrum, the Biozentrum Bio-EM lab, and the sciCORE scientific computing facility, all of University of Basel. We acknowledge the staff of beamlines X06DA and X06SA at the Paul Scherrer Institute, Villigen, Switzerland, for support with crystallographic data collection. We would like to thank DLS for beamtime and the staff of beamline B21 for SAXS data collection. MW and FM were supported by a Fellowship for Excellence from the Biozentrum Basel International PhD program. KB is supported by a Boehringer Ingelheim Fonds PhD Fellowship and has received support by a Biozentrum PhD Fellowship. LMC and MNH were supported by H2020 (ITN Healthage, grant agreement number 812830). This work was supported by the Swiss National Science Foundation (SNSF) via the National Center of Excellence in RNA and Disease (141735, 182880) to MNH and SNSF project and R'Equip funding to TM (179323, 177084).

## Additional information

### Funding

| Funder | Grant reference number | Author |
| --- | --- | --- |
| Swiss National Science Foundation | 179323 | Timm Maier |
| Swiss National Science Foundation | 177084 | Timm Maier |
| Swiss National Science Foundation | 141735 | Michael N Hall |
| Swiss National Science Foundation | 182880 | Michael N Hall |
| Horizon 2020 | 812830 | Louise-Marie Craigie Michael N Hall |

The funders had no role in study design, data collection and interpretation, or the decision to submit the work for publication.

### Author contributions

Matthias Wälchli, Writing – original draft, Writing – review and editing, designed experiments, expressed and purified proteins, prepared samples for cryo-EM, carried out data processing and structure modelling, analyzed data; Karolin Berneiser, Writing – review and editing, designed experiments, expressed and purified proteins, performed activity assays and analyzed data; Francesca Mangia, Writing – review and editing, expressed and purified proteins; Stefan Imseng, Writing – review and editing, cloned mTORC2 and contributed to data analysis; Louise-Marie Craigie, Writing – review and editing, designed experiments; Edward Stuttfeld, Writing – review and editing, established the mTORC2 purification procedure; Michael N Hall, Writing – review and editing; Timm Maier, Writing – original draft, Writing – review and editing, designed experiments

### Author ORCIDs

Matthias Wälchli http://orcid.org/0000-0001-9354-3079
Karolin Berneiser http://orcid.org/0000-0002-0963-8559
Francesca Mangia http://orcid.org/0000-0001-5430-397X
Stefan Imseng http://orcid.org/0000-0002-0248-0795
Edward Stuttfeld http://orcid.org/0000-0003-3932-9076
Michael N Hall http://orcid.org/0000-0002-2998-0757
Timm Maier http://orcid.org/0000-0002-7459-1363

### Decision letter and Author response

Decision letter https://doi.org/10.7554/eLife.70871.sa1

Author response https://doi.org/10.7554/eLife.70871.sa2

## Additional files

### Supplementary files
• Supplementary file 1. Cryo-EM data collection and refinement statistics of DEP domain-containing mammalian target of rapamycin (mTOR) interacting protein (DEPTOR)-mTOR complex 2 (mTORC2) complex.

• Supplementary file 2. Cryo-EM data collection and refinement statistics of DEP domain-containing mammalian target of rapamycin (mTOR) interacting protein (DEPTOR)-mTOR complex 1 (mTORC1) complex.

• Supplementary file 3. X-ray data collection and refinement for DEP domain-containing mammalian target of rapamycin (mTOR) interacting protein (DEPTOR) DEP domain tandem (DEPt).

• Transparent reporting form

### Data availability
Structure factors and model coordinates for the DEPt X-ray crystal structure is deposited to the RCSB protein data bank with PDB ID 7PED. Cryo-EM reconstructions and model coordinates are deposited to the EMDB and PDB for the mTORC2:DEPTOR dimer (EMDB ID: EMD-13347, PDB ID: 7PE7), mTORC2:DEPTOR protomer (EMDB ID: EMD-13348, PDB ID: 7PE8), mTORC2:DEPTOR protomer:DEPt (EMDB ID: EMD-13349, PDB ID: 7PE9), mTORC1:DEPTOR dimer (EMDB ID: EMD-13350, PDB ID: 7PEA), mTORC1:DEPTOR protomer(EMDB ID: EMD-13351, PDB ID: 7PEB), mTORC1:DEPTOR protomer:DEPt (EMDB ID: EMD-13352, PDB ID: 7PEC).

The following dataset was generated:

| Author(s) | Year | Dataset title | Dataset URL | Database and Identifier |
|---|---|---|---|---|
| Waelchli M, Maier T | 2021 | cryo-EM structure of DEPTOR bound to human mTOR complex 2, overall refinement | https://www.rcsb.org/structure/7PE7 | RCSB Protein Data Bank, 7PE7 |
| Waelchli M, Maier T | 2021 | cryo-EM structure of DEPTOR bound to human mTOR complex 2, overall refinement | https://www.ebi.ac.uk/emdb/entry/EMD-13347 | Electron Microscopy Data Bank, EMD-13347 |
| Waelchli M, Maier T | 2021 | cryo-EM structure of DEPTOR bound to human mTOR complex 2, focussed on one protomer | https://www.rcsb.org/structure/7PE8 | RCSB Protein Data Bank, 7PE8 |
| Waelchli M, Maier T | 2021 | cryo-EM structure of DEPTOR bound to human mTOR complex 2, focussed on one protomer | https://www.ebi.ac.uk/emdb/entry/EMD-13348 | Electron Microscopy Data Bank, EMD-13348 |
| Waelchli M, Maier T | 2021 | cryo-EM structure of DEPTOR bound to human mTOR complex 2, DEPt-bound subset local refinement | https://www.rcsb.org/structure/7PE9 | RCSB Protein Data Bank, 7PE9 |
| Waelchli M, Maier T | 2021 | cryo-EM structure of DEPTOR bound to human mTOR complex 2, DEPt-bound subset local refinement | https://www.ebi.ac.uk/emdb/entry/EMD-13349 | Electron Microscopy Data Bank, EMD-13349 |

*Continued*

| Author(s) | Year | Dataset title | Dataset URL | Database and Identifier |
|---|---|---|---|---|
| Waelchli M, Maier T | 2021 | cryo-EM structure of DEPTOR bound to human mTOR complex 1, overall refinement | https://www.rcsb.org/structure/7PEA | RCSB Protein Data Bank, 7PEA |
| Waelchli M, Maier T | 2021 | cryo-EM structure of DEPTOR bound to human mTOR complex 1, overall refinement | https://www.ebi.ac.uk/emdb/entry/EMD-13350 | Electron Microscopy Data Bank, EMD-13350 |
| Waelchli M, Maier T | 2021 | cryo-EM structure of DEPTOR bound to human mTOR complex 1, focussed on one protomer | https://www.rcsb.org/structure/7PEB | RCSB Protein Data Bank, 7PEB |
| Waelchli M, Maier T | 2021 | cryo-EM structure of DEPTOR bound to human mTOR complex 1, focussed on one protomer | https://www.ebi.ac.uk/emdb/entry/EMD-13351 | Electron Microscopy Data Bank, EMD-13351 |
| Waelchli M, Maier T | 2021 | cryo-EM structure of DEPTOR bound to human mTOR complex 1, DEPt-bound subset local refinement | https://www.rcsb.org/structure/7PEC | RCSB Protein Data Bank, 7PEC |
| Waelchli M, Maier T | 2021 | cryo-EM structure of DEPTOR bound to human mTOR complex 1, DEPt-bound subset local refinement | https://www.ebi.ac.uk/emdb/entry/EMD-13352 | Electron Microscopy Data Bank, EMD-13352 |
| Waelchli M, Jakob R, Maier T | 2021 | DEPTOR DEP domain tandem (DEPt) | https://www.rcsb.org/structure/7PED | RCSB Protein Data Bank, 7PED |

The following previously published datasets were used:

| Author(s) | Year | Dataset title | Dataset URL | Database and Identifier |
|---|---|---|---|---|
| Pavletich NP, Yang H | 2017 | mTORC1 structure refined to 3.0 angstroms | https://www.rcsb.org/structure/6BCX | RCSB Protein Data Bank, 6BCX |
| Scaiola A, Mangia F, Imseng S, Boehringer D, Ban N, Maier T | 2021 | cryo-EM structure of human mTOR complex 2, overall refinement | https://www.rcsb.org/structure/6ZWM | RCSB Protein Data Bank, 6ZWM |

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
