## [Decision Letter]

**Acceptance summary:**

DEPTOR is a regulator of the central mTORC1 and mTORC2 kinase complexes that is both of general interest to biologists and has remained poorly understood despite many years of investigation. Two *eLife* manuscripts report new structural insights into DEPTOR's mechanism of action that will be of broad impact for cell biologists, kinase enzymologists and mTORC1/mTORC2 specialists.

**Decision letter after peer review:**

Congratulations, we are pleased to inform you that your article, "The interaction of the regulator DEPTOR with mTOR complexes 1 and 2", has been accepted for publication in *eLife*. Please prepare your revision taking the specific comments of the peer reviewers into account as well as your discussions with the authors of the companion manuscript from the Williams group.

Your article has been reviewed by three peer reviewers, one of whom is a member of our Board of Reviewing Editors, and the evaluation has been overseen by a Senior Editor.

*Reviewer #1:*

mTORC1/mTORC2 activities coordinate metabolism and other environmental inputs with growth and division. mTORC1/2 dysfunctions contribute to cancers, metabolic derangements, autoimmune and neurological disorders. DEPTOR is an endogenous protein modulator of mTORC1 and mTORC2 that is of general interest because of its complex and still poorly understood mechanism of action, its roles in human disease, and the hope that understanding how DEPTOR works will lead to new strategies for therapeutically tuning mTORC1/2 activities. DEPTOR itself is regulated via phosphorylation by mTOR complexes. DEPTOR is the only protein reported to inhibit both mTORC1/2 complexes, and its inhibitory mechanism has been unclear for many years.

In this study, the group of Tim Maier succeeded in determining two cryo-EM structures of full-length DEPTOR bound to mTORC1 and mTORC2. In addition, the authors determined an X-ray crystal structure of DEPTOR's isolated tandem DEP domain (DEPt) to guide the interpretation of the cryo-EM maps and associated mutagenesis studies. Together, the authors succeeded in providing new structure/function insights into the binding modes of the PDZ and DEPt domains to both mTORC1 and mTORC2. These structures offer a potential explanation for the kinase stimulating effects of the PDZ domain alone, versus the inhibitory effects of the full-length protein that emerge through the low-affinity, but avid interaction of the tandem DEP domain with both mTORC1 and mTORC2. Together, these structures and structure-guided mutants provide a new model for DEPTOR actions on mTORC1 and mTORC2. The authors propose that the DEPTOR PDZ domain binds with a higher affinity to "anchor" DEPTOR to mTORC1/2, followed by lower affinity binding by DEPTOR's characteristic DEP tandem domains to suppress allosteric activation of mTOR.

When considered alongside additional recent studies of DEPTOR's functional relationship with mTORC1, this study deepens our understanding by providing insight into how DEPTOR functions on both mTORC1 and mTORC2, especially the role of the tandem DEP domains.

This study should make an impactful contribution to structural biologists, kinase enzymologists, and cell biologists. The major outstanding issue that remains to be clarified by future structure/function studies concerns a putative substrate-like binding mode between DEPTOR and the FRB domain of mTORC1 that may occur at high DEPTOR concentrations.

1) How does DEPTOR's linker become a substrate for mTORC1 and mTORC2? cis or trans?

In both this study and a related pre-preprint by the group of Roger Williams (https://www.biorxiv.org/content/10.1101/2021.04.28.441853v1), weak, poorly resolved, and largely uninterpretable cryoEM density is observed in association with the FRB domain of mTORC1 only (the FRB is not available for binding in mTORC2 because of RICTOR binding). The density is assigned to the linker region of DEPTOR by the Williams group, but this interpretation appears to be at odds with the structures reported here which resolved the PDZ and DEPt domains of full-length DEPTOR bound to both mTORC1 and mTORC2. The structural models reported here offer an alternative explanation of why DEPTOR needs its linker region for its unusual partial inhibition of mTORC1: placement of the DEPT domain relative to the PDZ domain. The authors of this study propose that the "substrate-like" mode of binding to the FRB domain of mTORC1 is lower in affinity than the interfaces between the PDZ and DEPt domains described here--and that only at high local DEPTOR concentrations will FRB binding take place, enabling mTORC1 to phosphorylate DEPTOR. At the same time though, these authors write that "we consider any relevance of a PRAS40-like FRB-interaction as the core mechanism for high-affinity inhibition of activated mTOR complexes as highly unlikely." This reviewer is inclined to think the density seen near the FRB domain has been misinterpreted as belonging to DEPTOR's long linker. To clarify the issue, would it be possible to mutagenize the surface of the FRB to disrupt the PRAS40 substrate binding mode, and assay how these mutants alter DEPTOR binding and phosphorylation by mTORC1 in cis?

*Reviewer #2:*

In order to explain how DEPTOR regulates mTOR activity in both the mTORC1 and mTORC2 complexes, this manuscript provides new structural insights into how DEPTOR binds to mTOR and provides thoughtful analysis of how such binding fits with and helps to explain a number of previous observations published on this topic. In particular, the manuscript explains how multiple distinct interaction interfaces between DEPTOR and mTOR contribute to regulation of mTOR activity. Key aspects of the predictions arising from the structural studies were tested and validated by in vitro kinase assays with DEPTOR mutants designed to selectively uncouple specific DEPTOR domains from binding to mTOR. This work therefore advances understanding of how mTOR activity is regulated by DEPTOR and is important for both the mTOR field as well as for investigation of DEPTOR contributions to the growth of various cancers.

This is a reasonably straight forward effort to shed light on the complex problem of how DEPTOR regulates mTOR signaling. Structural efforts reveal DEPTOR bound to both mTORC1 and mTORC2. A two step model is proposed that incorporates higher affinity mTOR binding by the DEPTOR PDZ domain and lower affinity binding by the DEPt domain. This model supports reasonable speculation about how additional factors (such as phosphatidic acid) could provide modulation.

The manuscript is well written and the major claims of the authors are supported by the available data. Additional speculation is reasonable. My only suggestion is that the mutagenesis and in vitro kinase assays presented in Figure 3C require some additional clarification. More specifically, the rationale for each of the mutations could be more clearly defined by illustration of exactly where the affected residues reside within the proposed interfaces between DEPTOR and mTOR. Also, the precise identity of each of the mutations should be better defined in the figure and its legend as the current names for each mutant do not clearly describe exactly what has been altered.

*Reviewer #3:*

In this manuscript, Walchli et al. provide structural insights on how DEPTOR, a component of both mTORC1 and mTORC2 endowed with complex regulatory functions, binds to the two complexes. They identify a bipartite mode of binding where the PDZ domain of DEPTOR engages mTOR at the interface between the bridge and FAT domains, whereas the tandem DEP domain (DEPt) mainly binds to the FAT domain. While thought to be lower affinity than the PDZ-mTOR, the DEPt-mTOR interaction plays an inhibitory function on mTORC1 kinase activity, as shown by the effect of predicted destabilizing mutations at the DEPt-mTORC1 interface on substrate phosphorylation. Moreover, because neither the PDZ nor the DEPt bind to the kinase active site, and because DEPTOR-bound mTORC1 is in the inactive conformation, it is likely that the inhibitory mechanism is allosteric. Based on structural considerations, a second (low affinity) binding mode is proposed in which DEPTOR is phosphorylated by mTORC1 in its linker region, which would engage close to the kinase active site.

Overall, this paper provides valuable insights into the regulatory mechanisms of DEPTOR toward mTORC1, and is supported by high quality structural and biochemical data.

1. Figure 3C: the effects of the DEPt domain mutations on 4EBP1 phosphorylation should be supported by binding assays between these mutants and mTORC1.

2. Also, the same kinase assay in Figure 3C could be repeated under basal conditions (Rheb-free) to support the point that DEPt-dependent mTORC1 inhibition only occurs in the Rheb-activated state.

3. There are significant differences with the structural arrangement proposed in the Heimhalt paper (which is quoted here). In particular, the simultaneous binding of DEPTOR PDZ and long linker to mTOR in the Heimhalt structure is incompatible with the dual PDZ + DEPt binding proposed here. The authors should discuss possible reasons for these differences.

---

## [Author Response]

Reviewer #1:[…]This study should make an impactful contribution to structural biologists, kinase enzymologists, and cell biologists. The major outstanding issue that remains to be clarified by future structure/function studies concerns a putative substrate-like binding mode between DEPTOR and the FRB domain of mTORC1 that may occur at high DEPTOR concentrations.1) How does DEPTOR's linker become a substrate for mTORC1 and mTORC2? cis or trans? […] To clarify the issue, would it be possible to mutagenize the surface of the FRB to disrupt the PRAS40 substrate binding mode, and assay how these mutants alter DEPTOR binding and phosphorylation by mTORC1 in cis?

We have carefully considered this aspect but considered that it is difficult to mutagenize an extended interface in the FRB that has multiple roles in interacting with substrates and inhibitors without possible affecting also substrate turnover. In the accompanying paper, Williams et al. use NMR-based mapping as a method to delineate DEPTOR-linker:FRB interactions. We also note that in our electron density maps of mTOR DEPTOR complexes, we don’t observe any density in the TOS motif binding site, which would be indicative of the presence of copurified PRAS40.

Reviewer #2:[…] My only suggestion is that the mutagenesis and in vitro kinase assays presented in Figure 3C require some additional clarification. More specifically, the rationale for each of the mutations could be more clearly defined by illustration of exactly where the affected residues reside within the proposed interfaces between DEPTOR and mTOR. Also, the precise identity of each of the mutations should be better defined in the figure and its legend as the current names for each mutant do not clearly describe exactly what has been altered.

We have added a new supplementary figure (Figure 3 supplement 2) clearly indicating the location of all mutations in the respective interfaces of DEPt and PDZ domains with mTOR and have updated figure 3 and the figure caption to have clear names and descriptions for all mutants.

Reviewer #3:In this manuscript, Walchli et al. provide structural insights on how DEPTOR, a component of both mTORC1 and mTORC2 endowed with complex regulatory functions, binds to the two complexes. They identify a bipartite mode of binding where the PDZ domain of DEPTOR engages mTOR at the interface between the bridge and FAT domains, whereas the tandem DEP domain (DEPt) mainly binds to the FAT domain. While thought to be lower affinity than the PDZ-mTOR, the DEPt-mTOR interaction plays an inhibitory function on mTORC1 kinase activity, as shown by the effect of predicted destabilizing mutations at the DEPt-mTORC1 interface on substrate phosphorylation. Moreover, because neither the PDZ nor the DEPt bind to the kinase active site, and because DEPTOR-bound mTORC1 is in the inactive conformation, it is likely that the inhibitory mechanism is allosteric. Based on structural considerations, a second (low affinity) binding mode is proposed in which DEPTOR is phosphorylated by mTORC1 in its linker region, which would engage close to the kinase active site.Overall, this paper provides valuable insights into the regulatory mechanisms of DEPTOR toward mTORC1, and is supported by high quality structural and biochemical data.1. Figure 3C: the effects of the DEPt domain mutations on 4EBP1 phosphorylation should be supported by binding assays between these mutants and mTORC1.

We consider the activity assays provided here as more indicative of function then the binding assay. Particularly for the Dept domains, whose function is a main aspect of this figure, a binding assay appears difficult: The effect of the DEPt domain depends on anchoring by the PDZ domains. Due to the limited maximal concentrations achievable for DEPt and in particular mTORC1, detecting binding even for the nonmutant version of DEPt may not be easily achieved with standard methods.

2. Also, the same kinase assay in Figure 3C could be repeated under basal conditions (Rheb-free) to support the point that DEPt-dependent mTORC1 inhibition only occurs in the Rheb-activated state.

In our experimental design we have focused on the physiologically revelant state of activated mTORC1. Heimhalt et al. have carried out careful and reliable in-depth kinetic characterization also for non-activated mTORC1.

3. There are significant differences with the structural arrangement proposed in the Heimhalt paper (which is quoted here). In particular, the simultaneous binding of DEPTOR PDZ and long linker to mTOR in the Heimhalt structure is incompatible with the dual PDZ + DEPt binding proposed here. The authors should discuss possible reasons for these differences.

We have addressed this difference in direct exchange with the authors of Heimhalt et al., and agree that there are two possible modes of interaction of DEPTOR with mTORCs: A substrate-like interaction specific to mTORC1, which may involve interactions of the DEPTOR linker with the FRB domain, and a regulatory interaction with bound PDZ and DEPt domains. Even if only the higher affinity PDZ domain is bound to mTOR, some mTOR-phosphorylation sites in DEPTOR could sterically not reach the active site of the same mTOR molecule. We understand, that in the final versions of both manuscripts the two modes of interactions are discussed, small changes have been made in our manuscript in the discussion session to more clearly point out the existence of these two modes.